# Close to open—Factors that hinder and promote open science in ecology research and education

Christian B. Strømme[1]*, A. Kelly Lane[3], Aud H. Halbritter[1], Elizabeth Law[2,4], Chloe R. Nater[2], Erlend B. Nilsen[2], Grace D. Boutouli[3], Dagmar D. Egelkraut[1], Richard J. Telford[1], Vigdis Vandvik[1], Sehoya H. Cotner[1]

**1** Department of Biological Sciences, University of Bergen, Bergen, Norway, **2** Norwegian Institute for Nature Research, Trondheim, Norway, **3** Department of Biology Teaching and Learning, University of Minnesota, Minneapolis, Minnesota, United States of America, **4** Working Conservation Consulting, Fernie, BC, Canada

* christian.stromme@uib.no

**Data Availability Statement:** All data and code are shared openly on a GitHub public repository: https://github.com/christianstromme/ LivingNorway2020.

## Abstract

The Open Science (OS) movement is rapidly gaining traction among policy-makers, research funders, scientific journals and individual scientists. Despite these tendencies, the pace of implementing OS throughout the scientific process and across the scientific community remains slow. Thus, a better understanding of the conditions that affect OS engagement, and in particular, of how practitioners learn, use, conduct and share research openly can guide those seeking to implement OS more broadly. We surveyed participants at an OS workshop hosted by the Living Norway Ecological Data Network in 2020 to learn how they perceived OS and its importance in their research, supervision and teaching. Further, we wanted to know what OS practices they had encountered in their education and what they saw as hindering or helping their engagement with OS. The survey contained scaled-response and open-ended questions, allowing for a mixed-methods approach. We obtained survey responses from 60 out of 128 workshop participants (47%). Responses indicated that usage and sharing of open data and code, as well as open access publication, were the most frequent OS practices. Only a minority of respondents reported having encountered OS in their formal education. A majority also viewed OS as less important in their teaching than in their research and supervisory roles. The respondents' suggestions for what would facilitate greater OS engagement in the future included knowledge, guidelines, and resources, but also social and structural support. These are aspects that could be strengthened by promoting explicit implementation of OS practices in higher education and by nurturing a more inclusive and equitable OS culture. We argue that incorporating OS in teaching and learning of science can yield substantial benefits to the research community, student learning, and ultimately, to the wider societal objectives of science and higher education.

**Funding:** EBN 312085 The Norwegian Research Council https://www.forskningsradet.no/en/ The funders had no role in study design, data collection and analysis, decision to publish, or preparation of the manuscript. VV 287784 The Norwegian Research Council https://www.forskningsradet.no/en/ The funders had no role in study design, data collection and analysis, decision to publish, or preparation of the manuscript. VV 274831 The Norwegian Research Council https://www.forskningsradet.no/en/ The funders had no role in study design, data collection and analysis, decision to publish, or preparation of the manuscript. VV Fra Vugge til Grad (unnumbered) Olav Thon Foundation https://olavthonstiftelsen.no/english/ The funders had no role in study design, data collection and analysis, decision to publish, or preparation of the manuscript.

**Competing interests:** The authors have declared that no competing interests exist.

# Introduction

Open Science (OS) covers a broad set of principles and practices aimed at improving the quality, reproducibility, efficiency, accessibility and, ultimately, the societal impact of science [1]. OS is currently gaining traction across the sciences, a trend that is driven by networks of OS practitioners as well as by research funders and policymakers [2, 3]. Generally, the OS movement aims to make research methods, data, results, and products freely available to everyone. These objectives are formulated on the basis of numerous ideals, namely increasing the diverse and equitable access and contribution to knowledge; research innovation and efficiency through collaboration; quality and credibility through transparency; access to research platforms; new and efficient tools and services; and alternative metrics for assessing research contribution and impact [3, 4]. Together, the rapidly evolving OS principles and practices are expected to revolutionise how research is done and shared in the future [5].

The transition towards OS has required the development of necessary infrastructure, including platforms for collaboration and large-scale interactive databases, and is currently redefining the landscape of funding opportunities and publishing models (e.g. Plan S [6]; Peer Community in [7]). This transition is supported by developments in licensing, data and metadata standards, and by requirements for open publishing and data sharing set by research funding bodies. While many practitioners adopt a subset of OS principles and practices for idealistic or pragmatic purposes, institutional support is crucial for developing OS infrastructure and mainstreaming OS [8, 9]. Moreover, OS principles conceptualised by practitioners can be adopted and implemented by institutions, as exemplified by the inclusion of the FAIR guiding principles [10] by the European Open Science Cloud [11] and the newly adopted UNESCO Recommendation on Open Science [12]. Through such efforts, many of the key institutional, economic and infrastructure-related challenges in the transition to Open Science have been addressed.

Despite these developments, OS is not widely implemented across research communities, instead remaining largely confined to groups, networks or events involving committed OS practitioners, as well as separate grassroots efforts that emphasise few or singular OS aspects [13, 14]. Thus, a major challenge for realising the potential of OS is to achieve a more widespread uptake across the scientific community, requiring a major cultural and behavioural shift among scientists. Such a transition is not straightforward since individual researchers vary in how they perceive OS in relation to their beliefs, skills and ability to reap benefits and bear costs [15]. For example, some practitioners may be motivated to adopt OS practices as a means to increase reproducibility and replicability in science, while others may be driven by ideals of free societal access to publicly funded knowledge. Further, individuals adopt OS differently based on personal background such as gender, nationality, and career stage [4]. Hence, there may be considerable variability in how OS is perceived, practised and promoted among individual researchers and between groups or networks.

These challenges call for more knowledge of how OS practices and principles are learned, understood, and transmitted in different parts of the scientific community. Such information can aid institutions and policymakers in mainstreaming OS across the sciences. We addressed these knowledge needs by surveying OS-related experiences and perceptions among attendees at an two-day international workshop dedicated to openness and transparency in applied ecology. This event was organised by the Living Norway Ecological Data Network [16], a peer-driven collaborative initiative originally established in 2019 to build better infrastructures for managing ecological data from Norwegian research institutions. Since then, this network has gradually grown to a more general-interest OS community for ecology in Norway and beyond. Living Norway is closely associated with the Norwegian participant node in the Global

Biodiversity Information Facility (GBIF [17]). The OS movement is still relatively young, and we saw the event as an opportunity for learning more about dedicated OS practitioners' thoughts on the role of OS in research, supervision, and teaching. For this purpose, we developed a digital survey for a mixed-methods approach, integrating quantitative and qualitative analyses, to specifically address the following research questions:

- How do OS practitioners in ecology perceive and define OS?

- Which OS aspects do practitioners in ecology interact with, and how frequently?

- What are the perceived benefits and risks for individual engagement in OS?

- Which OS aspects have practitioners in ecology encountered in their own formal education?

- How do ecologists engaged in OS as part of their research value OS in their teaching and supervision?

## Methods

### Colloquium

The 2nd International Living Norway Colloquium was a hybrid event held in October 2021. Participants could attend either in person or via a digital meeting platform. Across two days, participants could attend thematic sessions consisting of plenary talks, plenary and group discussions, and group assignments both in person and online (see S1 Table in S1 File for the full program).

### Survey development and administration

Our investigative team included individuals involved in organising the colloquium itself and associated collaborators. We met twice before the colloquium to clarify research questions, develop survey items and subject the items to talk-aloud refinement. This resulted in a questionnaire that we structured into three parts distributed to colloquium participants as follows: Part I three days before the event started, Part II at the end of the first day, and Part III after the second and final day of the event. We split the survey to distribute the effort of respondents taking the survey, to enable a focus on different themes, and to possibly infer whether participants' perceptions of OS evolved during the event.

### Survey structure and content

The questionnaire consisted of a combination of constrained choice, Likert-scale, and open-ended questions. Constrained choice questions were used to obtain background information such as degree, affiliation, gender, and familiarity with key OS aspects. Likert-scale response questions were related to experience with and perceived importance of different OS aspects to research, teaching, and supervision of students' thesis work. Open-ended questions asked colloquium participants to define OS and describe what had helped and hindered their OS engagement. Thus, the combination of question types allowed both qualitative and quantitative lines of inquiry. To link the submitted responses across the three parts, survey participants were asked to provide their email as an identifier with the assurance that this identifier would be removed within two weeks after the event. All three parts of the survey began with an information page where respondents were asked for consent, while also informing respondents of the study, handling of personal information and their options for withdrawing consent (see S1 File for details). The complete survey questionnaire is provided in S2, S3 Tables in S1 File.

## Data management

We uploaded and distributed the electronic survey using SurveyXact (Ramboll, Denmark) and this service provider collected the survey responses. Email addresses submitted by respondents were deleted and replaced with anonymous individual identifiers within two weeks, and remaining data were uploaded to a GitHub repository together with our R code for data cleaning and analyses. We clarified management procedures for handling personal information with a Data Protection Officer at NSD (Norwegian centre for research data) and registered the project in the System for Risk and Compliance (RETTE) at the University of Bergen.

## Qualitative analysis

We established a team (GDB, SHC, and AKL) that coded the open-ended responses to the following three survey items: definitions of OS, what hinders-, and what helps individual OS engagement (see S2&S3 Tables in S1 File for the full questions). Specifically, the team subjected item responses to iterative rounds of inductive coding, beginning with a meeting to establish an initial codebook by reading all responses to a particular question and identifying common ideas [18]. Because the open-ended questions served as opportunities to elicit new ideas from this novel participant group, we did not begin with *a priori* categories but rather remained open to all ideas that were present in the data. For each question, after describing initial categories, at least two of the three coders independently coded all responses. The coders then met and discussed differences along with any additional ideas that did not fit into the original categories. The coders repeated this process until no new categories were needed to capture all relevant ideas. Final codebooks can be found in S4;S7, S8 Tables in S1 File. For the final coding step, at least two coders coded all responses an additional time using the final codebook and then met and came to consensus on the coding for all responses. All coding was done before any coders were aware of the results of the quantitative analyses (see below), and vice versa, to protect the integrity of the coding process and the quantitative analyses. We then discussed overarching themes that resulted from the coding analysis during writing of the manuscript. We have lightly edited some of the quotes reported for grammar and clarity.

## Quantitative analysis

In parallel to the qualitative analyses described above, we investigated whether engagement and perceptions of OS, as captured by scaled responses, could be related to experiences, affiliations and backgrounds of colloquium participants. A team (AH, EL and CBS) formulated a set of specific predictions related to our general research questions prior to accessing the data. These predictions were pre-registered to avoid post-hoc hypothesising (see file containing pre-registered predictions on the GitHub project repository [19]. We tested the following pre-registered predictions:

A. Researchers from academic institutions have engaged more in open science-related practices compared to other researchers, and more so for early-career researchers. The rationale behind this prediction is that we expect researchers with a university affiliation to have stronger incentives and weaker disincentives for OS engagement compared to non-academic research institutions who tend to be more involved in generating and managing proprietary data. Further, we expect higher familiarity with OS among early-career researchers as the associated practices and principles have been gaining traction in academia over the past few years.

B. Colloquium participants use open data more frequently than they contribute open data and code. We base this prediction on the observation that using data from existing

repositories requires less effort than generating and sharing your own data. Further, there are stronger incentives in the sciences for publishing research papers than for sharing data.

C.  Colloquium participants are more likely to use OS in supervision and teaching when they use it in their own research. Research-based education is a strong ideal and stated goal within universities, and we therefore expect practices and principles that scholars adopt in their research to also be included in their teaching and supervision.

D.  Colloquium participants that have been taught open science-related practices as part of their education are more likely to engage in those practices. We expect that adopting a practice is more likely if it has been part of one's formal research training than if one had to learn it independently later.

E.  Colloquium participants engaged in both research and education perceive OS practices to be more important in their research than in their teaching and supervision. Our rationale here is that open science ideals and practices are explicitly promoted and incentivised in relation to research, but less so in relation to teaching or supervision.

The statistical models tested in the quantitative analyses were based on these predictions, as described below. For each prediction, we also explored whether the outcomes varied with gender. We performed all quantitative analyses using R software (version 4.1.1 [20]) and the code is available in the GitHub project repository.

We conducted analyses of quantitative data as follows. We treated the scaled responses as ordinal data by using cumulative link models (clms, function *clm*) or cumulative link mixed models (clmms, function *clmm*) that includes random terms, both from the R package Ordinal [21]. Each global model, defined at the start of the model selection process, was based on the pre-registered predictions (see above) by including fixed terms, including interactions, that reflected the prediction. Some of the pre-registered predictions (numbered in the file on GitHub) yielded overlapping model structures (predictions referred to as 1.2 and 3.1 on GitHub), while some predictions could not be tested due to insufficient data (prediction 1.4). This information is reported in the code for quantitative analyses, accessible in the project repository on GitHub.

For analyses that included multiple responses from each survey participant, we included a random intercept term (individual, a numeric labelling variable) in each global model and used clmms. For each prediction, we used the function *dredge* in the MuMIn R package for selecting the final model [22]. For each model we determined the inclusion of fixed- and random terms based on the lowest AICc estimate. If multiple models had a difference of AIC $\leq 2$, we selected the model having the simpler structure as the final model. For global- and final model structure, see S5, S6 S9, S10 Tables in S1 File.

For both clms and clmms, final models were checked for violation of the proportional odds assumption, namely if any of the fixed term estimates varied with the response categories. As these fixed terms were nominal factors, nominal tests were conducted by using the function *nominal_test* for clms. As the same function does not apply to clmms, we performed nominal tests for these models through likelihood ratio tests, comparing the most parsimonious model and a model with the same structure except having the fixed factor in question specified as nominal in the formula, thus relaxing the proportional odds assumption. If these tests revealed violations of the proportional odds assumption, we relaxed this assumption in the final model using the term nominal for the fixed term in question. This setting allowed the regression parameters to vary between different levels of a given covariate for which the proportional odds assumption was violated.

## Results

### Surveyed respondents

Among 128 registered colloquium participants, 60 completed Part I, 51 completed Part II and 38 completed Part III. Four colloquium participants that responded to Parts II and III did not respond to Part I. The majority of respondents reported to be affiliated with universities (N = 37), followed by research institutes (N = 22), other affiliations (N = 5) and governmental agencies (N = 2). Among these, six respondents stated multiple affiliations. Further, the majority stated Norway as their country of work or study (N = 45), followed by countries outside the EU (N = 13) and within the EU (N = 8) (S1, S2 Figs in S1 File). Among these, four respondents stated multiple countries of affiliation. 27 participants identified as women, 32 as men, and 1 as non-binary. In statistical analyses where Gender was used as a fixed term, the latter category was omitted as a factor level as it was represented only by a single participant.

### How is OS perceived among practitioners in ecology?

We asked colloquium participants to respond to the following open-ended survey item:

*People define 'Open Science' in many ways, and it is a multi-faceted concept. We are interested in how you define Open Science, especially as it pertains to your own work.*

Several codes emerged from the participant responses (S4 Table in S1 File). Quantifying the occurrence of these codes allows us to infer shared as well as less common and less salient perceptions of the meaning of OS (Table 1). In the pre-event responses, *shared data* was the most frequently identified code in these definitions of OS, with the vast majority (50 out of 60 respondents) indicating that in OS, data are available for anyone to use (e.g., "Sharing published data and/or raw data in open repositories"). The other most frequent statements included *data availability/accessibility* (38), *sharing codes/methods* (36), *transparency* (28) or *open access publications* (24) (Table 1). For example, one respondent commented, "open science means open access to scientific publications and open sharing of data and code for analysis for scientific publications" (coded as *sharing codes/methods*, *sharing data*, and *open access publications*). Another shared that "Open science is transparent and repeatable. In ecology

**Table 1. Frequency with which emergent codes were identified in the survey responses in Parts I and III.** Codes are organized from most frequent ("sharing data") to least frequent ("relationship between OS and education"), in Part I.

| Code or category | Number of occurrences in sample responses (before workshop; n = 60) | Number of occurrences in sample responses (after workshop; n = 38) |
|---|---|---|
| Sharing data | 50 | 29 |
| Accessibility/Availability | 38 | 14 |
| Sharing codes/methods | 36 | 23 |
| Transparency | 28 | 19 |
| Open access publications | 24 | 12 |
| Replication/Reproducibility | 19 | 13 |
| FAIR principles | 11 | 6 |
| Responsible & available to the public | 9 | 10 |
| Data policies & practices | 7 | 9 |
| Inclusivity | 5 | 2 |
| Working collaboratively with peers or other stakeholders | 5 | 6 |
| Relationship between OS and Education | 3 | 4 |

there is a particular need for data sharing, i.e. giving colleagues access to raw data for repeating analyses and/or applying alternative methods to extract information from the empirical data" (*transparency*, *sharing data*, *replication/reproducibility*).

Very few (3 of 60 respondents) defined OS in a way that specifically referenced education, or a relationship between OS and education, suggesting that it was not a common idea in this population. In all three mentions, education was listed as one of many facets of OS. For example:

"Open science is the ideal of free and accessible availability for everyone to all components of the scientific cycle. Open science entails open education, open research protocols, open methodologies, open data, open code, open data management and analysis, open research publication opportunities, open research readership opportunities, open data synthesis, open science-policy interface, and an open research funding and open science system, science policy and science management."

Similarly, few responses referenced inclusivity in their definitions. Those that did mention inclusivity described it as an important reason for OS. For example, one respondent shared:

"I would define Open Science, in the immediate sense, as a way of conducting research in a manner that is transparent (i.e. showcasing/sharing how you are doing your research), however this should extend past just sharing your research and it should also include creating and cultivating a research environment that is open and inclusive to all. Within my work the aspect of reproducibility (particularly code-sharing) is something that I place particular value on."

## Which OS aspects do practitioners interact with, and how frequently?

Among the 60 respondents to Part I of the survey, 49 were engaged in primary research. For prediction A, we found strong evidence that respondents with an academic affiliation had interacted with OS practices more frequently than those with other affiliations ($P = 0.003$) (Fig 1, S5 Table in S1 File). The frequency of interaction with OS practices was also higher among men than women ($P = 0.004$) (S5 Table in S1 File). However, we did not find evidence for higher OS engagement among early-career researchers, as the term was discarded in the model selection process (S5 Table in S1 File).

Respondents stated "Read open access publications" as the OS aspect that was most frequently engaged with, followed by "Used open code" and "Used open data" (Fig 2). For prediction B, we found strong evidence that respondents used open data and code more frequently than they shared data and code ($P < 0.001$), and men did so more frequently than women ($P = 0.002$) (S5 Table in S1 File).

## What are the perceived benefits and risks for individual engagement in OS?

Colloquium participants were asked, via open-ended questions, to share what had hindered and helped them to engage in OS. We developed separate codebooks for the "hinders" (S7 Table in S1 File) and "helps" (S8 Table in S1 File) responses. Of the 60 responses to the "hinders" item, many reported *lack of guidelines* (n = 15), *lack of time* (15), or *insufficient knowledge* (15) as barriers for engaging in OS (Table 2). One respondent exemplifies these sentiments (along with fear of critique) with the following:

"lack of familiarity with relevant online platforms, software, methods. . .perception that the landscape of the above tools changes very quickly, and keeping up is a big time commitment. . .fear of doing it wrong."

Seven colloquium participants wrote either *nothing* or *N/A*.

Of the 60 responses to the "help" item, 20 indicated *social support* as something that helped them to engage in OS (Table 3). Twenty also cited *resource availability*. One respondent covered both of these codes saying:

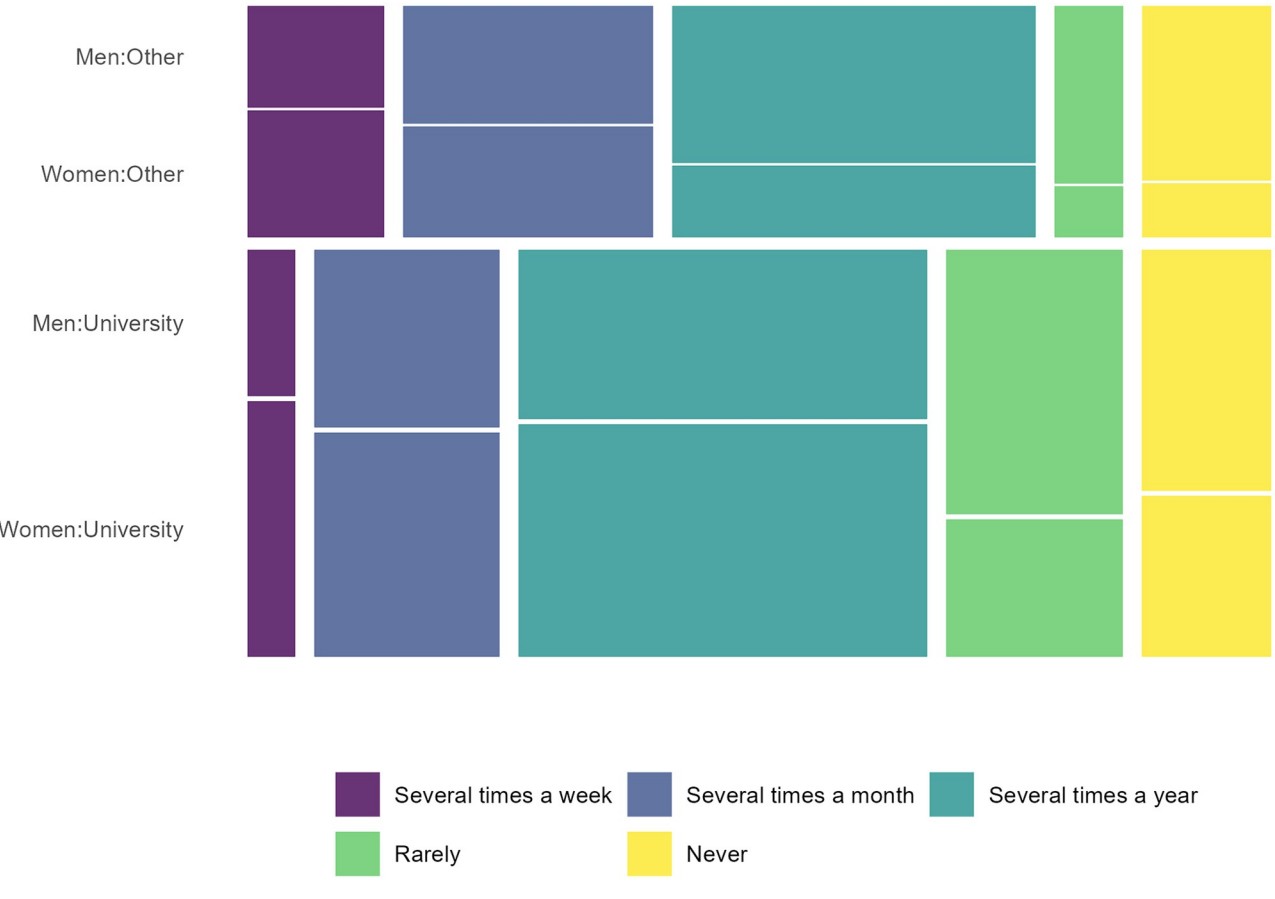

**Fig 1. Frequency of engagement with open science aspects according to the subset of colloquium participants engaged in research and responding to survey Part I (N = 36) by affiliation and gender.** Height of tiles corresponds to the number of participants for the respective categories of participants, width of tiles corresponds to frequency for each scale category.

"Abundance of open-source software, preprint servers, sci-hub (what a gem this is!), github, abundance of data repositories, support from colleagues."

Many of the perceived hindrances thus reflect the opposite of what is perceived to help, and vice versa.

## Which OS aspects have practitioners encountered in their own formal education?

About half of the surveyed colloquium participants (N = 29) stated that they had not encountered OS principles or practices in their own formal education (S3-S6 Figs in S1 File). Those participants that had encountered OS in their formal education reported the following:*read open access literature* (N = 21), followed by *used open code* (N = 19), *used open data* (N = 17), *shared open data* (N = 15), *published results or papers openly* (N = 15), *shared own code openly* (N = 13), *outreach/science communication* (N = 13),*been taught principles of research reproducibility* (N = 12),*used open-access online interactive learning resources* (N = 11),*been taught principles of research transparency* (N = 7), and *open peer review* (N = 4).

We predicted that colloquium participants that had encountered OS in their own education were more likely to engage with it in their research (Prediction D). We found evidence for participants that had encountered OS as part of their education used ($P = 0.027$) and shared open

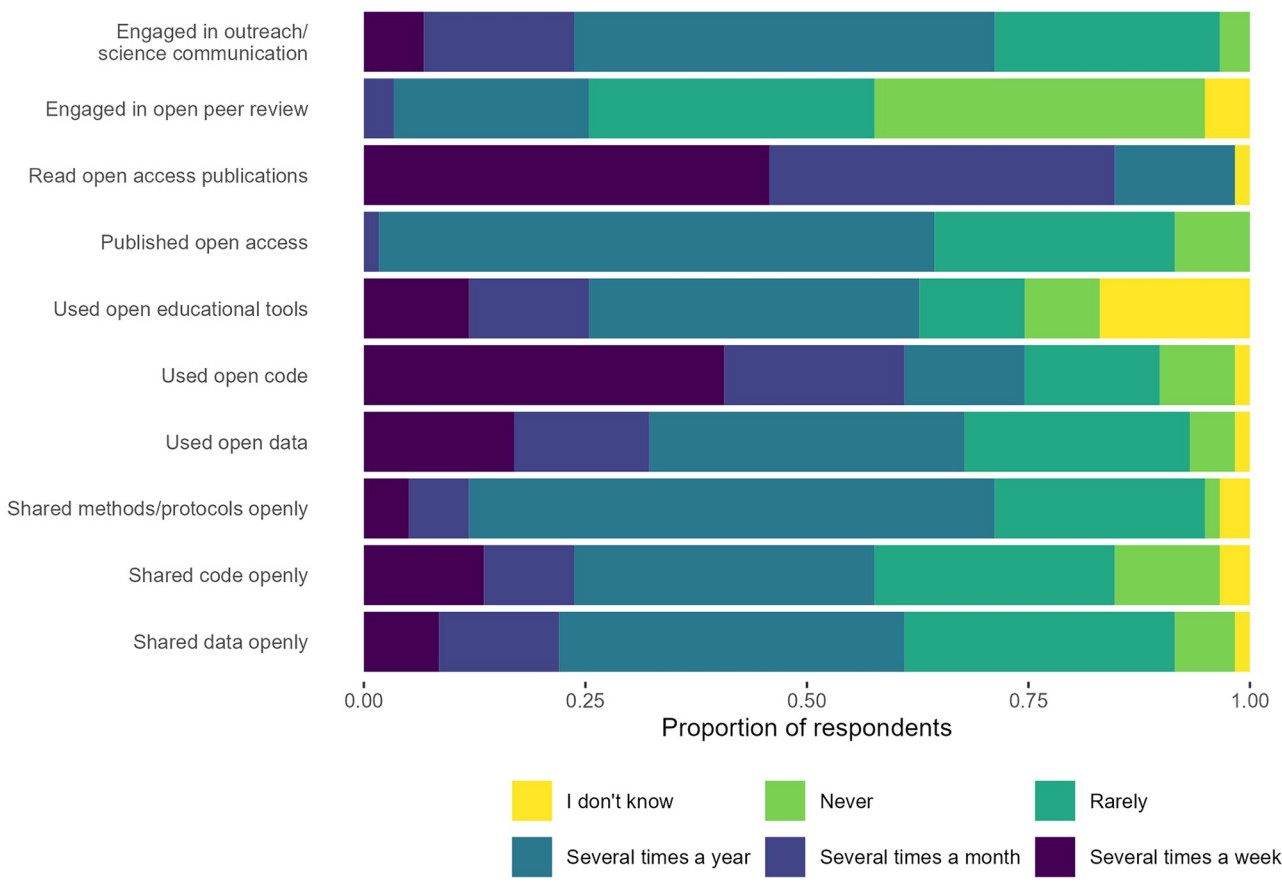

**Fig 2. Colloquium participants' stated frequency of engagement with different OS practices.**

code ($P = 0.023$), and used open educational tools ($P = 0.012$) more often than participants who had not encountered OS in their education. For these respondents, we found evidence that men shared ($P = 0.017$) and had used ($P = 0.028$) open data more frequently than women (S6 Table in S1 File).

**Table 2. Frequency with which emergent codes were identified for the responses to "what hinders your engagement in OS?" prompt.** Codes are organized from most frequent ("Lack of guidelines") to least frequent ("Fear of critique").

| Code | Number of occurrences in sample responses (Part I; n = 60) |
|---|---|
| Lack of Guidlines | 15 |
| Time | 15 |
| Insufficient knowledge | 15 |
| Collaborators not using OS | 13 |
| Other/Vague | 12 |
| Cost | 11 |
| Nothing | 7 |
| More work | 5 |
| Insufficient incentives | 5 |
| Legal concerns | 4 |
| Want to get credit | 3 |
| Fear of critique | 3 |

**Table 3. Frequency with which emergent codes were identified for the responses to "what helps your engagement in OS?" prompt.** Codes are organized from most frequent ("Social support") to least frequent ("Money").

| Code | Number of occurrences in sample responses (Part I; n = 60) |
|---|---|
| Social support | 20 |
| Resource availability | 20 |
| Intrinsic motivation | 13 |
| Structural support | 13 |
| Having knowledge | 8 |
| Other/Vague | 6 |
| Prior success with OS | 4 |
| Money | 4 |

## How do OS practitioners involved in higher education value OS in teaching and supervision of their students?

We found strong evidence that participants viewed OS as less important in teaching than in research ($P < 0.001$), but our data did not reveal any differences in perceived importance of OS between supervision and research (Fig 3; S9 Table in S1 File).

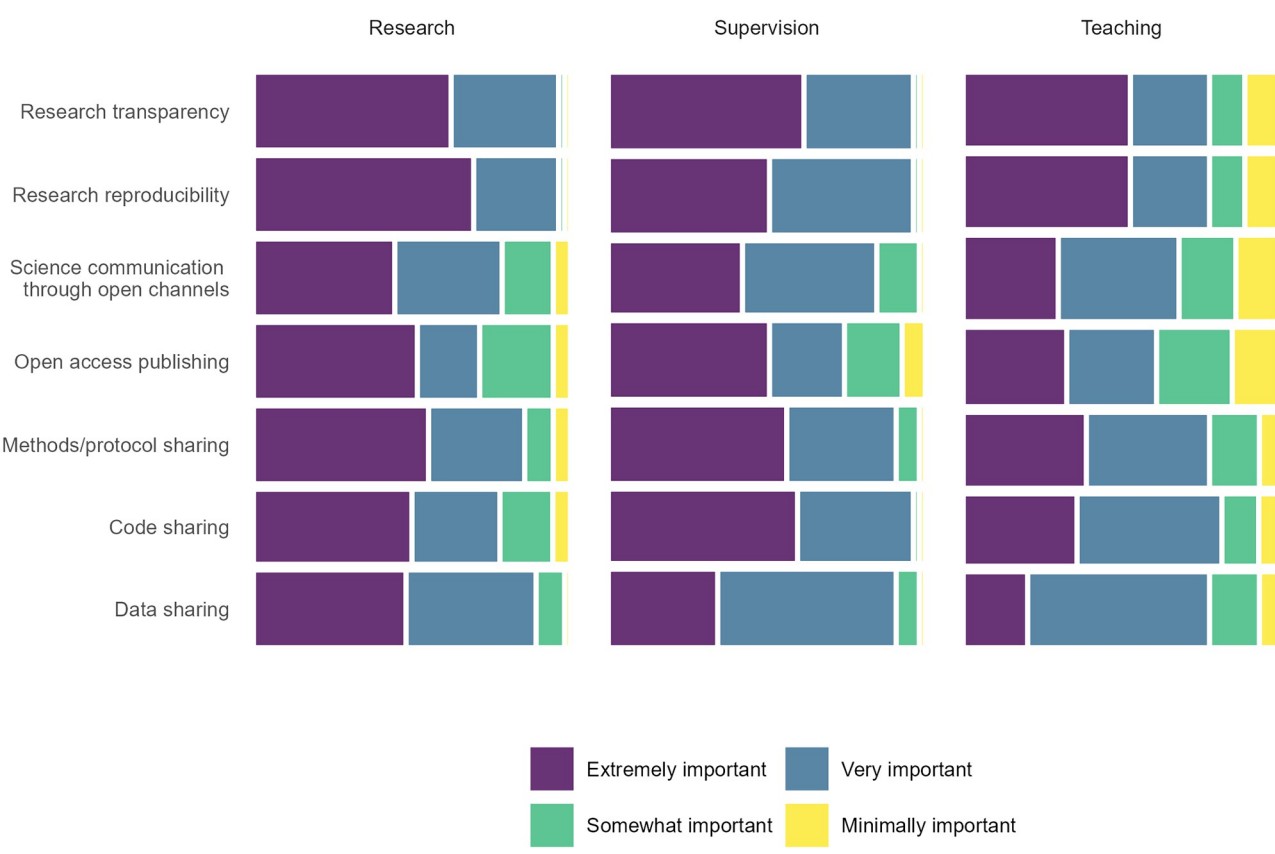

**Fig 3. Perceived importance of open science for research, teaching and supervision according to the subset of colloquium participants engaged in research and responding to survey Parts I and II (N = 26).** Width of tiles corresponds to the number of participants involved in the respective activities.

## Discussion

The data we received from surveyed participants at the Living Norway Ecological Data Network Colloquium 2020 illustrate how OS can be understood, applied and promoted in a network of dedicated OS practitioners. By combining qualitative and quantitative analyses to survey data, we identified two salient ideas, namely 1) that OS was mainly understood and practised in terms of shared and accessible data, code, transparency and publications, and 2) that the extent of OS engagement among respondents was linked to experience and perceptions of support. First, the respondents' own definitions of OS and the stated frequencies of engagement in related practices revealed that they commonly associated OS with *sharing data and code openly* and *open access publishing*. Second, an OS supportive community, knowledge, and resources were consistently seen as effective in promoting and broadening OS engagement, whereas the lack of these aspects were stated as hindrances. Third, respondents that had encountered these practices in their education reported more frequent engagement. Taken together, these results suggest that building an OS supportive community spanning both research and education can aid wider-scale implementation of OS. However, it is also clear from our data that this potential may not be evident to educators currently engaging in OS, considering that such practices were deemed less relevant in their context of teaching than in their research and supervision.

Since the main themes of the colloquium were Open Science and transparent data management, we asked the surveyed participants what they thought hindered and helped their individual engagement in OS. For the perceived hindrances, respondents most frequently identified *knowledge*, *time* and *lack of guidelines*. On the question of what had helped them engage with OS, they reported, *social support* and *resource availability*. Respondents also indicated that they used open code more frequently if they had experience with that activity in their own education. Taken together, these findings suggest that resources, a supportive community, and arenas for gaining experience and training are all necessary and helpful to enhance OS uptake.

A variety of existing examples and recommendations from available literature can guide the enhancement of OS via teaching and learning. Project EDDIE (Environmental Data-Driven Inquiry and Explorations) is a collaboration between disciplinary and educational researchers that has produced learning resources for students in a variety of natural science disciplines [23]. These are applications of active learning methods where students engage in inquiry and quantitative reasoning using large datasets. Other examples are the International Plant Functional Traits Courses that offer training in trait-based ecology through field campaigns focusing explicitly on data collection and data management grounded in FAIR open science practices. There, students plan and conduct reproducible fieldwork, manage data and publish data papers [24, 25]. Educators seeking to develop courses or learning resources where students engage with OS can get support for implementing OS in teaching and learning from workshops, courses and online tutorials (i.e. FOSTER Open Science [26]; the Open Science School [27]). In particular, Bossu & Heck have surveyed available literature on OS in teaching and learning while also providing advice on this topic [28]. We suggest that such adoptions of OS in undergraduate teaching can significantly leverage student engagement and learning [29], and thus speed up the implementation of OS in the wider community. Moreover, early engagement in OS can benefit students who consider research as a possible career option [24, 30, 31].

Increased uptake of OS in research and education can be promoted at different organisational levels: at grassroots level by practitioners, at the institutional level, or through research or higher education policy. We argue that a successful cultural transition towards OS depends

on an interplay across these levels. Considering the lower emphasis of OS in the context of teaching and learning among our respondents, it would be unrealistic to expect that grassroots practitioners alone can promote widespread implementation of OS across higher education. Rather, this transition is more likely to occur if aided by initiatives at the institutional and policy levels. While policies and sector-wide incentives are needed, mechanisms for structural and community support should be enacted at the institutional level and adapted to different academic cultures. Ultimately, scholars experienced with OS can offer the necessary know-how for the contexts in which learning happens.

While grassroots promotion of OS occurs both locally and globally (i.e. [32]), there have also been numerous institutional and political efforts, for example within the EU. The League of European Research Universities (LERU) has recommended universities to "integrate Open Science concepts, thinking, and its practical applications in educational and skills development programmes" [33]. In addition, the European Open Science Cloud, stemming from the European Commission Single Market strategy [11], aims at developing a "Web of FAIR Data and services" [34]. Despite these initiatives, implementations at the institutional level within the EU are still limited [14]. European universities surveyed in 2019 indicated research publications and attracting external funding as the most important activities for promoting research careers, while OS activities were considered of low importance [35]. Meanwhile, Utrecht University provides examples of how academic institutions can implement OS across the range of academic activities. The Utrecht University Open Science Platform was formulated by the institution itself and includes experienced practitioners in related formal processes. Further, the university has abandoned the impact factor when hiring and promoting staff while instead favouring OS-related activity and achievements [36]. It is likely that the state-level political context favours these efforts, considering that the Netherlands has a political commitment from 2013 to achieve 100% open access for publicly funded research publications [37]. Although publishing open access represents a limited aspect of OS, such policies constitute external pressures that may elicit change in affected institutions.

Despite clear benefits, implementing OS in research and higher education can also be accompanied by caveats. As OS is promoted by a variety of agents, the lack of a unifying definition gives room for diverse interpretations or even scepticism towards OS [38, 39]. Further, our data are in accordance with literature suggesting both practical, ethical and social barriers to OS engagement, such as lacking the required skills or support, or concerns with trade-offs pertaining to data sharing [40]. Our data further indicate that women engage less with OS than men, particularly in terms of using and sharing data and code, possibly reflecting lower confidence in their OS-related skills [41]. These results suggest a strong case for complementing practical support and incentives with mechanisms for social support around OS, as such mechanisms are generally known to disproportionately benefit women and marginalised groups [42]. Our data further suggests that those that are most involved already have access to social support networks, as our respondents pointed most frequently towards social support (alongside resource availability) when asked about what had helped their OS engagement. This may suggest that not only the existence of social support networks, but also equitable access to participate needs to be explicitly targeted to further enhance the implementation of OS across research and education.

In conclusion, our study provides insights into how OS is being and could be understood, applied and promoted within a community of practitioners. Respondents seemed to understand and practice OS mainly in terms of providing and/or re-using data and code in addition to open access publishing, but were less aware of how OS can support and promote quality science and education more broadly. Further, statements pertaining to what helps and hinders individual engagement in OS revealed aspects that can be addressed effectively through

building a collaborative and inclusive OS science culture, including through higher education and post-graduate training. Even though we can expect variation across the broader scientific communities, our results point at issues that deserve consideration. In particular, the differential emphasis of OS in research vs. teaching reflects a prolonged schism in academia where these two scholarly activities are typically regulated by dissimilar mechanisms. Therefore, implementing OS holistically in both research and higher education offers a unique opportunity to bring teaching and research closer together, ultimately advancing knowledge and its applications to the most pressing challenges of our time.

## Supporting information

**S1 File. Supporting information.**
(PDF)

## Author Contributions

**Conceptualization:** Christian B. Strømme, A. Kelly Lane, Elizabeth Law, Chloe R. Nater, Erlend B. Nilsen, Dagmar D. Egelkraut, Richard J. Telford, Sehoya H. Cotner.

**Data curation:** Christian B. Strømme, Aud H. Halbritter, Sehoya H. Cotner.

**Formal analysis:** Christian B. Strømme, A. Kelly Lane, Aud H. Halbritter, Elizabeth Law, Grace D. Boutouli, Richard J. Telford, Sehoya H. Cotner.

**Funding acquisition:** Erlend B. Nilsen, Vigdis Vandvik.

**Investigation:** Christian B. Strømme, A. Kelly Lane, Aud H. Halbritter, Elizabeth Law, Chloe R. Nater, Grace D. Boutouli, Vigdis Vandvik, Sehoya H. Cotner.

**Methodology:** Christian B. Strømme, A. Kelly Lane, Aud H. Halbritter, Elizabeth Law, Chloe R. Nater, Dagmar D. Egelkraut, Richard J. Telford, Vigdis Vandvik, Sehoya H. Cotner.

**Project administration:** Christian B. Strømme, Dagmar D. Egelkraut, Vigdis Vandvik.

**Resources:** Vigdis Vandvik.

**Software:** Christian B. Strømme, Aud H. Halbritter.

**Supervision:** Christian B. Strømme, Vigdis Vandvik.

**Visualization:** Christian B. Strømme, Aud H. Halbritter, Elizabeth Law, Sehoya H. Cotner.

**Writing – original draft:** Christian B. Strømme, A. Kelly Lane, Aud H. Halbritter, Elizabeth Law, Chloe R. Nater, Erlend B. Nilsen, Grace D. Boutouli, Vigdis Vandvik, Sehoya H. Cotner.

**Writing – review & editing:** Christian B. Strømme, A. Kelly Lane, Aud H. Halbritter, Elizabeth Law, Chloe R. Nater, Erlend B. Nilsen, Grace D. Boutouli, Richard J. Telford, Vigdis Vandvik, Sehoya H. Cotner.

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
