## [Decision Letter · Decision Letter 0]

26 Sep 2022

PONE-D-22-17298

CLOSE TO OPEN - FACTORS THAT HINDER AND PROMOTE OPEN SCIENCE IN ECOLOGY RESEARCH AND EDUCATION

PLOS ONE

Dear Dr. Strømme,

Thank you for submitting your manuscript to PLOS ONE. After careful consideration, we feel that it has merit but does not fully meet PLOS ONE’s publication criteria as it currently stands. Therefore, we invite you to submit a revised version of the manuscript that addresses the points raised during the review process.

Both reviewers agree on the merits of your paper. I would recommend to consider the few minor point raised by Reviewer #1  before the paper is finally accepted for publication.

We look forward to receiving your revised manuscript.

Kind regards,

Alberto Baccini, Ph.D.

Academic Editor

PLOS ONE

Journal Requirements:

Reviewers' comments:

Reviewer's Responses to Questions

**Comments to the Author**

1. Is the manuscript technically sound, and do the data support the conclusions?

Reviewer #1: Partly

Reviewer #2: Yes

2. Has the statistical analysis been performed appropriately and rigorously? 

Reviewer #1: Yes

Reviewer #2: Yes

3. Have the authors made all data underlying the findings in their manuscript fully available?

Reviewer #1: Yes

Reviewer #2: Yes

4. Is the manuscript presented in an intelligible fashion and written in standard English?

Reviewer #1: Yes

Reviewer #2: Yes

5. Review Comments to the Author

Reviewer #1: The work deals with a study based on a survey of a community of ecologists who attended the OS workshop hosted by the Living Norway Ecological Data Network in 2020 to learn how they perceived OS and its importance in their research, supervision and teaching. The survey (split in phases) is based on an extremely small number of respondents (at most about 60). The survey is conducted in a professional manner by adopting both quantitative and qualitative indicators. The extremely small number of data points makes it inconsistent from the statistical point of view. Even the changes in responses shown in Tab.1 are based on small datasets, thus making the differences between table columns marginally significant. I have troubles in understanding who such a work could be of interest, given the small size of the sample studied. However, PLOSONE seems to include into its scopes even very partial reports such as this study: "We evaluate submitted manuscripts on the basis of methodological rigor and high ethical standards, regardless of perceived novelty. (from PLOSONE web page) " .

Therefore, I find this work sufficiently consistent with PLOSONE standards and I recommend not to reject it.

A few minor points should be addressed prior to acceptance:

1) There is a well known correlation between OS and research assessment (RA). RA in some Countries is seen as the most important factor against OS. Almost no mention is made of RA in both study and paper (not even mention to S.F.DORA or the new European research evaluation initiatives). Only one sentence is shown about the Univ of Utrecht experience, that is interesting but definitely partial and not representative of the EU scenario. Text should be added describing the rationale of such choice of not discussing RA.

2) Throughout the paper references are shown in brackets instead as regular references. Please move them to the References section. Examples of references to be moved are in lines 17, 48, 53, 137, 364.

Reviewer #2: I recommend to accept the manuscript in this version.

The authors correctly address the comments made by previous reviewers, as a result the submitted manuscript is technically sound, the applied methodology is appropriate, clearly and extensively presented, the conclusions are rigorously built on results.

The manuscript is well presented and written in standard English.

6. PLOS authors have the option to publish the peer review history of their article (what does this mean?). If published, this will include your full peer review and any attached files.

Reviewer #1: **Yes: **Stefano Bianco - Senior Researcher - Laboratori Nazionali di Frascati dell' Istituto Nazionale di Fisica Nucleare.

Reviewer #2: **Yes: **Matteo Lascialfari

---

## [Author Response · Author response to Decision Letter 0]

31 Oct 2022

We are very grateful for the helpful comments, as they pointed to needed edits in the previously submitted manuscript. Please find the point-by-point replies and additional edits with line references below.

Reviewer #1

1) We agree with Reviewer 1 that this is an interesting aspect, as it can potentially impact institutional support for open science (OS). Though such impacts would ultimately manifest on the level of individual actions and motivations, we did not survey perceptions of research assessment or its connection to OS in explicit terms through the survey we developed. Further, our concern is that including this aspect post-hoc would be too speculative, especially since we did not identify the idea of research assessment across the responses to the survey items on helping and hindering individual OS engagement. Even so, we agree on the need to better describe the EU scenario in this respect. We have added a statement on the prevailing emphasis of research papers and external funding for individual careers as indicated by findings from a Europe-wide survey in 2019 (lines 385-388, reference at line 534). Thus, the passage on the Utrecht University that now follows the added sentence is an example of how research assessment can be adjusted for promoting OS.

2) Agreed. We have moved website references (found at lines 17, 47, 52, 134, 359 and 360) to the reference list (corresponding lines 447, 450, 477, 480, 486, 507 and 510) as suggested.

Additional changes

Line 388: We replaced “Even so” with “Meanwhile” to increase flow between the anteceding (newly inserted) sentence and this sentence.

---

## [Editor Report · Decision Letter 1]

15 Nov 2022

CLOSE TO OPEN - FACTORS THAT HINDER AND PROMOTE OPEN SCIENCE IN ECOLOGY RESEARCH AND EDUCATION

PONE-D-22-17298R1

Dear Dr. Strømme,

We’re pleased to inform you that your manuscript has been judged scientifically suitable for publication and will be formally accepted for publication once it meets all outstanding technical requirements.

Kind regards,

Alberto Baccini, Ph.D.

Academic Editor

PLOS ONE
---

## [Editor Report · Acceptance letter]

12 Dec 2022

PONE-D-22-17298R1 

CLOSE TO OPEN - FACTORS THAT HINDER AND PROMOTE OPEN SCIENCE IN ECOLOGY RESEARCH AND EDUCATION 

Dear Dr. Strømme:

I'm pleased to inform you that your manuscript has been deemed suitable for publication in PLOS ONE. Congratulations! Your manuscript is now with our production department. 

Kind regards, 

on behalf of

Prof. Alberto Baccini 

Academic Editor

PLOS ONE